# An Interface Setup Optimization Method Using a Throughput Estimation Model for Concurrently Communicating Access Points in a Wireless Local Area Network

**DOI:** 10.3390/s23146367

**Published:** 2023-07-13

**Authors:** Fatema Akhter, Nobuo Funabiki, Ei Ei Htet, Bin Wu, Dezheng Kong, Shihao Fang

**Affiliations:** Graduate School of Natural Science and Technology, Okayama University, Okayama 700-8530, Japan; fatema@s.okayama-u.ac.jp (F.A.);

**Keywords:** wireless local area network, signal-to-interference ratio, interface setup optimization, throughput estimation model, channel bonding

## Abstract

The *IEEE 802.11 wireless local-area network (WLAN)* has been deployed around the globe as a major Internet access medium due to its low cost and high flexibility and capacity. Unfortunately, dense wireless networks can suffer from poor performance due to high levels of radio interference resulting from adjoining *access points (APs)*. To address this problem, we studied the *AP transmission power optimization method*, which selects the maximum or minimum power supplied to each AP so that the average *signal-to-interference ratio (SIR)* among the concurrently communicating APs is maximized.However, this method requires measurements of *receiving signal strength (RSS)* under all the possible combinations of powers. It may need intolerable loads and time as the number of APs increases. It also only considers the use of *channel bonding (CB)*, although *non-CB* sometimes achieves higher performance under high levels of interference. In this paper, we present an *AP interface setup optimization method* using the *throughput estimation model* for concurrently communicating APs. The proposed method selects *CB* or *non-CB* in addition to the maximum or minimum power for each AP. This model approach avoids expensive costs of RSS measurements under a number of combinations. To estimate the RSS at an AP from another AP or a host, the model needs the distance and the obstacles between them, such as walls. Then, by calculating the estimated RSS with the model and calculating the SIR from them, the AP interface setups for a lot of APs in a large-scale wireless network can be optimized on a computer in a very short time. For evaluation, we conducted extensive experiments using *Raspberry Pi* for APs and *Linux PCs* for hosts under 12 network topologies in three buildings at Okayama University, Japan, and Jatiya Kabi Kazi Nazrul Islam University, Bangladesh. The results confirm that the proposed method selects the best AP interface setup with the highest total throughput in any topology.

## 1. Introduction

The *IEEE 802.11 wireless local area network (WLAN)* has been deployed around the world as a major Internet access medium due to its simple installation, the low cost of devices, and its flexible extensions [1,2,3,4,5]. In a WLAN, a user is connected to an *access point (AP)* through a wireless medium to the Internet access service. Then, the connection capacity and the coverage area of the Internet service can be enhanced by installing new APs in the service field. As a result, a lot of APs are often installed randomly in service fields with default channel and power settings. Unplanned AP deployments result in dense WLAN environments, particularly in highly populated areas [6].   Figure 1 illustrates the dense WLAN deployment example.

With dense WLAN deployment, a user often suffers from poor network performance caused by interference among the radio signals for data transmission between APs and hosts using the same or similar radio frequencies that have overlapping spectra. This interference problem may be mitigated by allocating the non-interfered orthogonal channels to the APs [7,8,9]. However, the number of orthogonal channels is limited in the IEEE 802.11 WLAN. In the popular 2.4 GHz band, this number is limited to two for *channel bonding (CB)* channels and four for *non-CB* channels. It is noted that a *CB* channel is often used to enhance the transmission capacity in a WLAN by bonding two adjacent channels into one channel.

As another way to reduce the interference in the dense WLANs, a reduction in the transmission power of the AP should be considered. Low transmission power can shorten the transmission rage and make the interfered signal weak. However, it can also decrease the data transmission capacity of the link and the coverage area of the AP. Therefore, it is crucial to set the proper transmission power for each AP, considering the relative positions of the APs and the hosts in the network field.

To address the abovementioned problem, we previously proposed the *AP transmission power optimization method* for concurrently communicating APs in a WLAN. After assigning the orthogonal channels to minimize the interference, this method selects either the maximum or minimum transmission power to each AP such that the average *signal-to-interference ratio (SIR)* among the APs is maximized [10,11].

However, in this method, the necessary *received signal strength (RSS)* for calculating the SIR needs to be measured under the possible combinations of transmission powers of the APs. They include the RSS of the *target signal* from its associated host and the RSS of the *interference signals* from other Wi-Fi devices. The measurements may result in intolerable loads and time for the user when the number of APs increases. Furthermore, only the CB was considered in the previous method, although it was observed that the use of non-CB channels for some APs sometimes offers a higher throughput when the APs are very closely located in a dense WLAN. A non-CB channel can decrease spectrum overlapping among the APs and increase the number of orthogonal channels, which can make wireless links less susceptible to interference.

In this paper, we propose an *AP interface setup optimization method* using the *throughput estimation model* [12,13] for concurrently communicating APs. The method selects either the CB or non-CB, in addition to the selection of either the maximum or minimum transmission power to each AP. This model approach is adopted to avoid the costs of RSS measurements under this increasing number of combinations. Using the throughput estimation model, the necessary RSS to calculate the SIR under all combinations of the CB/non-CB and the maximum/minimum power is instantly estimated. After estimating the SIR, the best combination for the highest SIR is selected as the best AP interface setup.

For evaluations of the proposal, we conducted extensive experiments using *Raspberry Pi* for APs and *Linux PCs* for hosts under 12 network topologies in three buildings at Okayama University in Japan and at Jatiya Kabi Kazi Nazrul Islam University in Bangladesh. The results confirm that the proposed method selects the best AP interface setup of the CB/non-CB and the maximum/minimum transmission power selections that offers the highest total throughput in any topology.

The novelty of the proposed method is that the AP interface setup, including the selection of CB or non-CB and the selection of the maximum or minimum transmission power is optimized by the throughput estimation model simulations without conducting RSS measurement experiments under multiple combinations. The throughput estimation model is a simple two-stage model that can be easily implemented and tuned. To estimate the RSS at an AP from another AP or a host, the model needs the distance and the obstacles between them, such as walls. Then, by calculating the estimated RSS with the model and calculating the SIR accordingly, the AP interface setups for a lot of APs in a large-scale wireless network can be optimized on a computer in a very short time.

The weak point of the proposed method may be the accuracy of the throughput estimation model. However, it is only used to estimate the RSS, not the throughput, and calculate the SIR from the RSS. Thus, the result is not sensitive to the model accuracy. The impact of the model accuracy of the proposed method will be investigated in future studies.

The rest of this paper is organized as follows. Section 2 introduces related works in the literature. Section 3 reviews our previous work. Section 4 presents AP interface optimization using the throughput estimation model. Section 5 presents the experiment setup for evaluations. Section 6, Section 7, and Section 8.1 show the experimental results. Finally, Section 9 concludes this paper with directions for future work.

## 2. Related Works

In this section, we introduce some related works in the literature on transmission power optimizations in a WLAN. They address the joint channel and power assignment in dense WLANs.

In [14], Wu et al. proposed a joint channel allocation and power control scheme to minimize interference and improve the throughput. First, channels are allocated to the APs based on the principle that neighboring APs should have large channel spacing. Then, the power is adjusted according to the *signal-to-interference-noise ratio (SINR)*, ranging from the lowest power value to the default power value. However, only simulation results are presented, and only *non-CB* channels are used.

In [15], Kachroo et al. proposed a combined channel assignment and power optimization method to reduce interference. First, the optimal channel assignment is determined while keeping the other parameters such as the power and the position to constant. Then, the transmission power of each AP is optimized, taking the coverage area threshold into consideration. Again, only 20 MHz *non-CB* channels were used in simulations.

In [16] Garcia et al. proposed a heuristic algorithm for determining the optimal channel and power transmission configuration for the APs within a network. The signal-to-interference noise ratio (SINR) is considered as the metric for optimization. For SINR, an overlapping factor is assumed for channel spacing, and the average data rate is provided in accordance with SINR. However, no real measurement was provided to substantiate the proposal other than simulations.

In [9], Tewari et al. proposed a joint transmission power and *partially overlapping channel (POC)* assignment algorithm to maximize the network performance in dense WLANs. The authors considered only *non-CB* POCs, and the effectiveness was verified in simulations only.

In [17], Shitara et al. proposed a transmission power control scheme using an indicator that is issued from a neighbor AP. When the channel occupancy rate increases, the AP issues the indicator. The other APs adjust transmission powers based on the previous actions when they receive it. The effectiveness was verified only in simulations.

In [18], Daldoul et al. introduced the power constraints and the impacts on data rates in *IEEE802.11n/ac* protocols. A rate-ordering scheme called *MinstrelHT* is defined to select the best data rate to improve the performance. The effectiveness was verified only in simulations.

In [19], Zhao et al. proposed a joint power control and channel allocation method based on the reinforcement learning algorithm that combines the statistical channel state information to reduce the interference. An event-driven strategy is introduced to trigger the learning process and reacquire the optimal strategy. Only 20 MHz *non-CB* channels were considered in simulations.

In [20], Girmay et al. proposed a joint mode selection, channel allocation, and power control algorithm based on particle swarm optimization (PSO) to maximize the overall throughput. The mixed-integer nonlinear problem (MINP) is utilized to reduce the interference while ensuring the minimum data rate requirements for Wi-Fi users. However, only simulations were used to evaluate the proposal.

In [21], Garroppo et al. proposed an efficient technique for energy efficiency in WLANs. It switches off the powers of some APs and controls the transmission powers when the user activity is low. However, they only considered off-peak hours of Internet usage to optimize energy consumption. Most researchers evaluate the effectiveness of their proposals using only non-CB channels and simulations. On the other hand, the proposed method leverages both non-CB and CB channels, and the effectiveness is evaluated in both real testbed experiments and simulations. Some existing approaches increase the average or total network throughput by assigning either proper channel numbers or transmission powers to APs, while others use both. The proposed method selects the channel type and the transmission power for each AP simultaneously based on the average SIR. Table 1 compares the proposal and with related works in terms of relevant implementation issues.

## 3. Review of Previous Studies

In this section, we review our previous studies of the *AP transmission power optimization method* [11] and the *throughput estimation model* [12,13].

### 3.1. AP Transmission Power Optimization Method

First, we review the *AP transmission power optimization method*. The proposed method only considers either the minimum or maximum transmission power of an AP, although a modern AP supports a wide range of transmission power levels. However, our limitation of the power selection comes from our prior work reported in [22]. In this study, we measured the throughput using the testbed system when the AP transmission power was gradually changed from the minimum to the maximum, including medium values in various topologies. Then, we found that the throughput was highest when we selected either the maximum or minimum power at each AP in any topology. The best selection is different from the AP in each topology. Therefore, we studied the method of selecting the best transmission power for each AP in the given topology and found that the power selection resulting in the largest SIR in the topology results in the highest overall throughput. In this study, we propose a method of selecting the best transmission power by using the SIR estimated by the throughput estimation model.

#### 3.1.1. Significance of SIR

In a WLAN, the network performance can be maximized by properly setting up the interface, considering the capacity, the interference, and the coverage area [17,23]. The *signal-to-interference ratio (SIR)* is the metric used to assess the quality of a wireless communication link. SIR can characterize both the link capacity and the interference by taking the ratio between the *received signal strength (RSS)* and the interfered signals in the targeted device. A higher SIR suggests higher network performance, as reported in [24,25].

#### 3.1.2. Method Procedure

The procedure of the method is described here for three concurrently communicating APs using *CB* channels, where each AP is associated with one host. The Linux commands and bash scripts for the testbed implementation of this method are described in Appendix A.

Assign either the maximum ($P_{max}$) or minimum ($P_{min}$) transmission power to each AP. There are eight power combinations for the three APs.For each power combination, measure the following *received signal strength (RSS)* at the APs:$RSS_{H_{i},AP_{j}}$: RSS of the signal from host $H_{i}$ at $AP_{j}$ for i,j=1,2,3;$RSS_{AP_{i},AP_{j}}$: RSS of the signal from $AP_{i}$ at $AP_{j}$ for i,j=1,2,3;$RSS_{AP_{x},AP_{j}}$: RSS of the signal from an unknown AP in another WLAN at $AP_{j}$ for i,j=1,2,3.Convert the measured RSS from dBm to mW using the following equation:
(1)$$RSS_{mW}=1mW\times 10^{\left(RSS_{dBm}/10\right)}.$$
where:$RSS_{dBm}$ represents the RSS in Decibel-Milliwatt units (dBm); and$RSS_{mW}$ represents the RSS in Milliwatt units (mW).Calculate the SIR of each AP ($AP_{i}$) and $SIR_{AP_{i}}$ using the following equation:
(2)$$\begin{array}{ccc} & SIR_{AP_{i}}= & \frac{RSS_{H_{i},AP_{i}}}{\sum_{j=1,j\neq i}^{3}RSS_{H_{i},AP_{j}}+\sum_{j=1,j\neq i}^{3}RSS_{H_{j},AP_{i}}+\sum_{j=1,j\neq i}^{3}RSS_{AP_{j},AP_{i}}+\sum_{x\neq i}RSS_{AP_{x},AP_{i}}}. \end{array}$$Calculate the average SIR ($SIR_{avg}$) using the following equation:
(3)$$SIR_{avg}=\frac{1}{3}\left(SIR_{AP_{1}}+SIR_{AP_{2}}+SIR_{AP_{3}}\right).$$
where $SIR_{AP_{1}}$, $SIR_{AP_{2}}$, and $SIR_{AP_{3}}$ are the SIR of $AP_{1}$, $AP_{2}$, and $AP_{3}$, respectively.Find the power combination that has the highest average SIR among all power combinations, and assign the corresponding powers to the APs.

#### 3.1.3. Limitations

In this *AP transmission power optimization method*, two limitations should be pointed out. The first limitation is the sole use of the CB channel for any AP, since 40 MHz CB channels basically provide higher throughputs than 20 MHz non-CB channels. However, CB channels can make APs more susceptible to interference due to fewer non-interfered channels than non-CB channels. It has been found that in busy WLAN environments where several APs are communicating concurrently in the same field, non-CB channels may deliver higher performance than CB channels for APs. Thus, both CB and non-CB channels should be properly used for APs.

The second limitation is the exponential increase in the RSS measurements with the number of APs. In this method, RSS measurement is necessary for all the power combinations of the APs to obtain SIR for them, which is not suitable for practical applications. If the CB/non-CB channel assignment is additionally considered in addition to the power selection, the measurement loads are further increased. Thus, approaches other than measurement should be adopted to avoid the loads of RSS measurements.

### 3.2. Throughput Estimation Model

Next, we review the *throughput estimation model*. The throughput estimation model has two equations to estimate the throughput between a source node (AP) and a destination node (host). First, it estimates the *receiving signal strength* (*RSS*) at the host by using the *log distance path loss model*. Then, it converts the estimated *RSS* into the corresponding throughput using the *sigmoid function*.

The *RSS* ($RSS_{d}$ (dBm)) at the host is estimated as follows:
(4)$$RSS_{d}=P_{1}-10\alpha \log_{10}d-\sum_{k}n_{k}W_{k}$$
where $P_{1}$ represents the signal strength at 1m from the AP (source) for no obstacles, α is the path loss exponent, *d* (m) represents the link distance from the AP, $n_{k}$ is the number of type-*k* walls along the path between the AP and the host, and $W_{k}$ is the signal attenuation factor (dBm) for the type-*k* wall in the environment.The throughput (TP, Mbps) of a link between the AP and the host is calculated based on $RSS_{d}$ as follows:
(5)$$TP=\frac{a}{1+e^{-(\frac{(RSS_{d}+120)-b}{c})}}$$
where *a*, *b*, and *c* are the constant parameters of the sigmoid function that are to be tuned.

## 4. AP Interface Setup Optimization Method

In this section, we present the *AP interface optimization method* using the *throughput estimation model*.

### 4.1. Solutions to Limitations

In the proposed method, the *CB*/*non-CB* selection and the *throughput estimation model* are newly introduced to address the limitations of the previous method discussed in Section 3.1.3.

First, the channel type selections of *CB* or *non-CB* to the APs in addition to *transmission powers* are newly considered in the interface setup optimization method to maximize the total throughput by reducing interference among them. When *non-CB* channels are selected, more orthogonal channels can be assigned to the APs.

Second, the *throughput estimation model* is used to estimate the required *RSS* to calculate *SIR* instead of measuring it using the real devices. This model approach can substantially reduce the necessary time to optimize the AP interface setup.

### 4.2. Procedure

Figure 2 shows the flow of the AP interface optimization method. The procedure for *N* APs (N=3) is described as follows:Enumerate all the possible combinations of the CB/non-CB channel and the transmission power *(channel type and power)* for *N* APs. For one AP, four *(channel type and power)* combinations exist: (CB,max), (CB,min), (nonCB,max), and (nonCB,min). Thus, there are $4^{N}$*(channel type and power)* combinations for *N* APs;Select one *(channel type and power)* combination and estimate the necessary RSS ($RSS_{H_{i},AP_{j}}$, $RSS_{AP_{i},AP_{j}}$, or $RSS_{AP_{x},AP_{j}}$) by assigning them to the corresponding APs using the *throughput estimation model*;Convert the unit of the estimated RSS from dBm to mW using Equation (Equation 1);Calculate the SIR of the individual AP using Equation (Equation 2) and the average SIR using Equation (Equation 3);When the average SIR is not calculated for some *(channel type and power)* combinations, go back to step 2;Find the *(channel type, and power)* combination that has the highest average SIR among all the *(channel type and power)* combinations and assign the corresponding channels and the transmission powers to the APs.

## 5. Experimental Setups

In this section, we discuss the setups used in our experiments for evaluation.

### 5.1. Running Platform

Table 2 shows the PC platform used to run the proposed method.

### 5.2. Model Parameters

Table 3 shows the parameter values in the *throughput estimation model*. ${P_{1}}_{max}$ and ${P_{1}}_{min}$ represent the value of $P_{1}$, which is the signal strength at 1 m from the AP when the maximum and minimum transmission power are assigned to the AP, respectively.

### 5.3. Devices and Software for Measurements

In our experiments, *Raspberry Pi* [26] with a USB wireless NIC adapter is adopted for the AP by running *Host Access Point Daemon (hostapd)* [27]. The built-in NIC adapter of *Raspberry Pi* is used for the 20 MHz *non-CB* channel. The USB wireless NIC adapter is used for the 40 MHz *CB* channel, since the built-in NIC adapter of the adopted *Raspberry Pi* does not support the CB. A laptop PC with a *Linux* operating system (OS) is used for the server and the host. The 2.4 GHz frequency band is used for experiments.

To measure the throughput of a wireless link, TCP downlink traffic from the server to the host is generated using *iperf* [28] with a 477 kbyte TCP window and an 8 kbyte buffer. The server is connected to the AP by a wire. TCP downlink traffic is common in WLANs, since users often download data from servers on the Internet using TCP through web site accesses. The Linux tool *iw* [29] is used to measure RSS at the APs and to change the transmission power of the AP. Table 4 shows the specifications of the devices and software used in the experiments.

### 5.4. Network Topologies and Fields

To evaluate the proposed AP interface setup optimization method through experiments, 13 network topologies in three network fields are considered. Table 5 shows the locations of the APs and the hosts in the field for each topology. For any topology, the AP and its associated host are located in the same room as the usual situation in a WLAN.

*Topologies 1–6* are made on the 3rd floor of the *Engineering Building #2* at Okayama University (*OU-Eng*), Japan. In this field, there are eight rooms with two different room sizes of 7 m × 6 m and 3.5 m × 6 m and one corridor with a size of 30 m ×2.3 m. Six topologies are designed to consider different interference levels in experiments.

*Topologies 7–10* are made on the 2nd floor of the *Graduate School of Natural Science and Technology Building* at Okayama University (*OU-Grad*), Japan. In this field, there is one large room of 17 m × 16 m in size and seven rooms with various sizes. Among them, the large room and the three rooms with a size of 9 m × 6 m are used. Four network topologies are designed to consider different interference levels.

*Topologies 11–14* are made on the 2nd floor of the *Science Building* at Jatiya Kabi Kazi Nazrul Islam University (*JU-Sci*), Bangladesh. In this field, there are six rooms with two different room sizes of 8 m × 7 m and 4 m × 7 m and one corridor with a size of 32 m ×2.3 m. Four topologies are designed to consider different interference levels in experiments.

Two APs and hosts are used in topologies 1, 2, 7, and 8; three APs and hosts are used in topologies 3, 4, 5, 6, 9, and 10; five APs and hosts are used in topologies 11 and 12; and 10 and 15 APs and hosts are used in topologies 13 and 14, respectively. Each AP is connected to one server PC by a wired cable, and one host PC is connected by wireless signal.Traffic is generated and downloaded to the host at the server using *iperf*.

## 6. Experimental Results in *OU-Eng*

In this section, we present experimental results for two APs and three APs at *OU-Eng*.

### 6.1. Results for Two APs

This section presents the experiment results for two APs in two interference scenarios of high and low interference, where all the possible (channel type and power) combinations of two APs are considered.

#### 6.1.1. Case 1: High Interference

In *topology 1*, the two APs are placed closely together in the same room (D307), as shown in  Figure 3. The distance between $AP_{1}$ and $AP_{2}$ is 1.5 m. This topology can result in strong interference between the APs.

Table 6 shows the estimated SIR (e.SIR) obtained from the RSS esimated by the model, the average SIR (a.SIR) from the RSS measured in the experiment, and the summation of the measured throughputs of the two links or the total throughput (t.thp) for each of the four transmission power combinations and the three CB/non-CB channel combinations for the two APs. In relation to the transmission power, only the high or low transmission powers appear to produce optimal throughput and are represented here by H and L, respectively.

The results indicate that the estimated SIR can identify the optimal combination of power and channel type that provides the highest total throughput. Thus, the efficacy of the proposed method is confirmed for this topology. In this topology, the combination of two CB channels (1+5) and (9+13) with (H,L) powers actually provides the highest total throughput. This result is justified based on the following observations:The use of two CB channels can allow for the utilization of the full capacity of the frequency spectrum, maximizing the total throughput;Since the APs are located in the same room, one with the minimum power can reduce the cochannel interference and increase the throughput;Due to network congestion in the D307 environment, the low power in $AP_{2}$ can maximize the total throughput in this network field.

#### 6.1.2. Case 2: Low Interference

In *topology 2*, $AP_{1}$ is located in D307, and $AP_{2}$ is located in the corridor in front of D301, as shown in  Figure 4. The distance between the APs is significantly greater than the distance in topology 1. $AP_{1}$ is separated from $AP_{2}$ by several walls. Thus, this topology has less interference than topology 1. It is noted that the multipath effect is considered for e.SIR to estimate the RSS of the interfered signal [12] because the direct signal along the *line of sight (LOS)* between the APs becomes weak due to the presence of the four walls. The selection of the indirect signal can reduce the number of boundaries and their impacts on signal strength.

The results presented in Table 7 once again confirm that the highest SIR estimated by this model yields the highest overall throughput. In *topology 2*, the two CBs with the (L,L) powers have the maximum throughput, as explained by the following observations:The usage of CB channels at the APs makes full use of the frequency spectrum;The low powers at both APs can help to avoid interference with other APs in this environment and improve the performance;The close distance between the AP and its associated host also contributes to the power selection reasoning.

### 6.2. Results for 3 APs

This section presents the experimental results in the four topologies for all possible (channel type and power) combinations of three APs.

#### 6.2.1. Case 3: Very High Interference

In *topology 3*, the three APs are located close together in the same room (D307), as shown in  Figure 5. The distance between $AP_{1}$ and $AP_{2}$ and that between $AP_{2}$ and $AP_{3}$ is 1.5 m. This topology can cause strong interference among the APs.

Table 8 shows the estimated SIR (e.SIR), average SIR (a.SIR), and total throughput (t.thp) for each of the eight transmission power combinations and the four CB/non-CB channel combinations for the three APs.

The results again confirm the correlation between the estimated SIR and the best combination of the power and the channel type, which offers the highest total throughput for a topology. Thus, the effectiveness of the proposed method is confirmed for three APs as well. In this topology, the combination of three *non-CB* channels using channels 1, 7, and 13 with (H,L,H) powers actually provides the highest total throughput, as explained by the following observations:The use of a CB channel at an AP can be interfered with by any CB/non-CB channel of another AP in this strong interference case. Thus, the use of CB channels decreases the total throughput;On the other hand, the uses of three non-CB distant channels is less susceptible to interference, which can increase the total throughput;Because $AP_{2}$ is located in the middle of $AP_{1}$ and $AP_{3}$, the maximum power of $AP_{1}$ and $AP_{3}$ and the minimum power of $AP_{2}$ can increase the transmission opportunities at $AP_{1}$ and $AP_{3}$ while reducing them at $AP_{2}$, which can increase the total throughput.

#### 6.2.2. Case 4: High Interference

In *topology 4*, $AP_{1}$ is located in D306, and $AP_{2}$ and $AP_{3}$ are located in D307; these rooms are separated by one wall, as shown in  Figure 6. The distance between $AP_{1}$ and $AP_{2}$ is 8m, and that between $AP_{1}$ and $AP_{3}$ is 9.5 m. This topology can cause moderate interference among the APs compared to *topology 1*.

The results presented in Table 9 show a similar trend, where the combination with the highest SIR estimated by the model offers the highest total throughput. In *topology 4*, the one-CB and two-non-CB combinations with the (L,H,H) powers provide the highest throughput, as explained by the following observations:The use of a CB channel at $AP_{2}$ and $AP_{3}$ can increase their transmission capacities;The use of distant non-CB channels at $AP_{2}$ and $AP_{3}$ can reduce the interference between them;The interference between $AP_{1}$ and the other APs is much smaller than that in *topology 1* due to the separating wall.

#### 6.2.3. Case 5: Low Interference

In *topology 5*, $AP_{1}$ is located in D307, $AP_{2}$ is located in the refresh corner, and $AP_{3}$ is located in the corridor in front of D301, as shown in Figure 7. The distances between the APs are much larger than those in the previous two topologies. $AP_{1}$ is separated from $AP_{2}$ and $AP_{3}$ by a wall, whereas there is no wall between $AP_{2}$ and $AP_{3}$. Thus, this topology has less interference than the previous two topologies.

The results presented in Table 10 follow a pattern in finding the best combination with the highest SIR estimated by the model, providing the highest total throughput. In *topology 3*, the combination of two CB channels and one non-CB channel with the (L,L,H) powers provides the highest total throughput. It is noted that channel 13 is assigned to $AP_{3}$ instead of channel 1 because channel 13 is less crowded in this field, as explained by the following observations:The interference between $AP_{1}$ and $AP_{2}$ is minimized due to the distance and the separating wall. Thus, the orthogonal CB channels with minimum power can increase the total throughput while reducing the interference;Any non-CB/CB channel at $AP_{3}$ can be interfered with by $AP_{1}$ or $AP_{2}$. Thus, the non-CB channel with maximum power at $AP_{3}$ can increase the total throughput by properly activating the CSMA/CA protocol against $AP_{2}$ while minimizing the interference against $AP_{1}$.

#### 6.2.4. Case 6: Very Low Interference

In *topology 6*, $AP_{1}$ is located in D308, $AP_{2}$ is located in D306, and $AP_{3}$ is located in D302. They are separated by at least two walls, as shown in  Figure 8. The distance between them is larger than that in the previous topologies. Thus, this topology has the lowest interference among the four topologies in *Engineering Building #2*.

The results presented in Table 11 recur in finding the combination with the highest SIR estimated by the model, offering the highest total throughput. In *topology 6*, the combination of three CB channels with the (H,H,H) powers provides the highest total throughput, as explained by the following observations:The interference among the APs is minimized due to the distances and the separating walls between the APs. Thus, the use of the orthogonal CB channels with the maximum transmission power can increase the total throughput.

## 7. Experimental Results in *OU-Grad*

In this section, we present the experimental results for two APs and three APs in the Graduate School Building, Okayama University, Japan.

### 7.1. Result for Two APs

This section presents the experimental results involving two APs with high and low interference, as previously presented, with consideration of all the possible (channel type and power) combinations of two APs.

#### 7.1.1. Case 7: High Interference

In *topology 7*, the two APs are in room F, as illustrated in  Figure 9. The distance between $AP_{1}$ and $AP_{2}$ is 1.5 m again. Although the room is larger than that in topology 1–6, there is strong interference between the APs due to their close proximity.

The results presented in Table 12 maintain that the combination with the highest SIR estimated by the model offers the highest total throughput. In *topology 7*, the combination of two CB channels with the (H,L) powers provides the highest total throughput, as in *topology 1* in *OU-Eng*.

#### 7.1.2. Case 8: Results for Low Interference

In *topology 8*, $AP_{1}$ and $AP_{2}$ are located in different rooms with separating walls between rooms *A* and *E* respectively, as shown in Figure 10. In this topology, $AP_{1}$ and $AP_{2}$ are three rooms apart. Thus, this topology is expected to have less interference than the previous topology in this building.

The results presented in Table 13 show that the combination with the highest SIR estimated by the model offers the highest total throughput. *Topology 8* is reminiscent of *topology 2* from *OU-Eng* in selecting the combination of two CB channels with the (L,L) powers, which provides the highest total throughput. Here, the multipath effect is also considered for e.SIR to estimate the RSS of the interfered signal.

### 7.2. Results for Three APs

This section presents experimental results for three AP networks while the interference varies.

#### 7.2.1. Case 9: High Interference

 Figure 11 illustrates the high-interference scenario for *topology 9* in *OU-Grad* with three APs. All three APs are located in the same room with a size of 17 m× 16 m. The distance between $AP_{1}$ and $AP_{2}$ and that between $AP_{2}$ and $AP_{3}$ is 1.5 m. Each AP is connected to one host with a 1 m distance. This topology can cause strong interference among the APs due to the close proximity of the APs.

The results presented in Table 14 are persistent in that the combination with the highest SIR estimated by the model offers the highest total throughput. In *topology 9*, the combination of three non-CB channels with the (H,L,H) powers provides the highest total throughput, as in *topology 1* in the other building.

#### 7.2.2. Case 10: Low Interference

In *topology 10*, $AP_{1}$, $AP_{2}$, and $AP_{3}$ are located in different rooms with separating walls between rooms *A*, *C*, and *G*, respectively, as shown in Figure 12. In this topology, $AP_{1}$ and $AP_{2}$ are relatively closer to each other compared with $AP_{3}$. Thus, this topology experiences less interference than the aforementioned topology.

The results presented in Table 15 confirm that the combination with the highest SIR estimated by the model offers the highest total throughput. In *topology 10*, the combination of two CB channels and one non-CB channel with the (L,H,H) powers provides the highest total throughput, as explained by the following observations:The interference between $AP_{2}$ and $AP_{3}$ is small due to the distance and the presence of multiple walls. Thus, the use of orthogonal CB channels with maximum power can increase the total throughput while reducing the interference.Any CB/non-CB channel at $AP_{1}$ can be interfered with at $AP_{2}$ or $AP_{3}$. Thus, non-CB channel 13 with minimum power at $AP_{1}$ can increase the total throughput by properly activating the CSMA/CA protocol against $AP_{3}$, which has small interference with $AP_{2}$.

## 8. Numerical Application in *JU-Sci*

In our experiments, the model-based method was found to discover the best channel-type–power combination. Now, we apply this method to a new network field with numerous APs where human assessment is difficult due to the large number of possible combinations.

### 8.1. Numerical Experiment with Five APs

This section presents the numerical experimental results of two topologies with every possible channel–power combination for five APs in *JU-Sci*. The distributions of APs and hosts are depicted in Figure 13 and Figure 14. Table 16 shows only the optimal channel–power combinations with the highest estimated SIR values. Channel–power combinations with lower estimated SIR values are not presented here due to the vast number of possibilities.

The results presented in Table 16 indicate that the proposed estimation model can find the channel–power combination that provides the highest SIR for each topology. The transmission power adjustment and the combination of CB and non-CB resulted in the highest SIR to optimize the network performance, as justified by the following observations:In *topology 11*, AP1 is somewhat distant from other APs. Therefore, the CB assignment with high power at AP1 does not cause interference with the other APs. The next three APs, (AP2-AP4) are relatively close to each other. Therefore, the non-CB assignment with low power can lower the interference. The last AP (AP5) is outside of the AP1 coverage and can function on the CB with high power to maximize the network throughput.In *topology 12*, AP1 and AP2 are closely located but far from the other three APs. Consequently, CB assignment to both APs does not cause strong interfere with the other APs. The low power in AP2 can reduce the interference with AP1. The remaining three APs (AP3-AP5) are positioned closely. Thus, non-CB assignment with high power to two end APs and CB assignment with *L* power to the middle AP can maximize the network throughput while minimizing interference.

### 8.2. Numerical Experiment with 10 APs

This section presents experimental results for 10 APs in *JU-Sci*. The distributions of APs and hosts are depicted in  Figure 15. Table 17 shows the optimal channel–power combinations with the highest estimated SIR values.

The topology induces high interference, since a large number of APs is placed in a small area. Nevertheless, from Table 17, we can see that the proposed method finds the best channel–power combination for this topology as well. Here, only the optimal channel–power combination with the highest estimated SIR is listed, as explained by the following observations:In *topology 13*, AP1 and AP2 are located in room 201 and are assigned non-CB channels 1 and 5, respectively. However, AP1 is assigned the maximum power, while AP2 is assigned minimum power. Since AP3 is in the corridor and is separated from the other APs by several walls, it is assigned the CB channel (9+13) with the maximum power;Next, AP4, located in the middle, can be interfered by APs from either side. Inevitably, AP4 is assigned non-CB channel 7 with maximum power to cover its connected host on the other side of a wall;AP5 and AP6, which are in the same room, perform best on non-CB channels 11 and 3 with minimum powers, respectively;AP7, which is the only one AP in room 204, is assigned to CB channel (1+5) with the minimum power to reduce the interference, since the adjoining APs are assigned to non-CB channels;AP8 can be interfered from both sides, whereas AP9 is interfered from one side only; these APs are assigned to non-CB channels 13 and 1, respectively. In terms of the transmission power, AP8, as a middle AP, is assigned the minimum power to reduce the interference. AP9 is assigned the maximum power, as it is located in the end room;The last AP, AP10, receives a similar assignment to that of AP3, since they are both located in the corridor, varying only in the channel (5+9).

### 8.3. Numerical Experiment with 15 APs

This section presents the experimental results for 15 APs in *JU-Sci*. The locations of the APs and the hosts are depicted in  Figure 16. In this topology, they cause high interference. Therefore, most of the APs are assigned to non-CB channels. Only two APs at the ends of the corridor are assigned to CB channels.

Table 18 shows the optimal channel–power combination with the highest estimated SIR. This topology has extremely high interference, since many APs are placed in a small field. Even in this very crowded environment, the proposed method finds the best channel-type–power combination for the optimal performance. The best combination can be explained by the following observations:AP5 and AP15 in the corridor are separated from the other APs by multiple walls and distance. Thus, they are assigned to CB channels (1+5) and (9+13), respectively, with the maximum power;The rest of the APs are assigned to non-CB channels, since they are closely located. The end APs, like AP1 and AP14, and the isolated APs, like AP6 and AP11, are assigned the maximum power. The remaining APs are assigned the minimum power to reduce the overall interference in the network.

Table 19 summarizes the experimental results presented in this paper. The results presented in Table 19 show that the proposed method works properly in all the network topologies in the three different buildings at different levels of interference.

It should be noted that the effectiveness of the proposal is verified by evaluating the estimated throughput relative to the measured throughput. However, comparisons with other methods are still worthy of further investigations in future work.

## 9. Conclusions

In this paper, we presented the *access-point (AP)* interface setup optimization method using the *throughput estimation model* for concurrently communicating APs in a wireless local area network (WLAN). The proposed method estimates the *signal-to-interference ratio (SIR)* for every combination of CB/non-CB channels and the maximum/minimum transmission power and selects that with the highest SIR. The experimental results obtained using two, three, and five APs in 12 different network topologies in three buildings with different interference levels confirm the validity of the proposal. In future works, we will examine the effect of channel assignment and apply the proposal to various network environments including different numbers of APs and hosts in a WLAN.

## Figures and Tables

**Figure 1 sensors-23-06367-f001:**
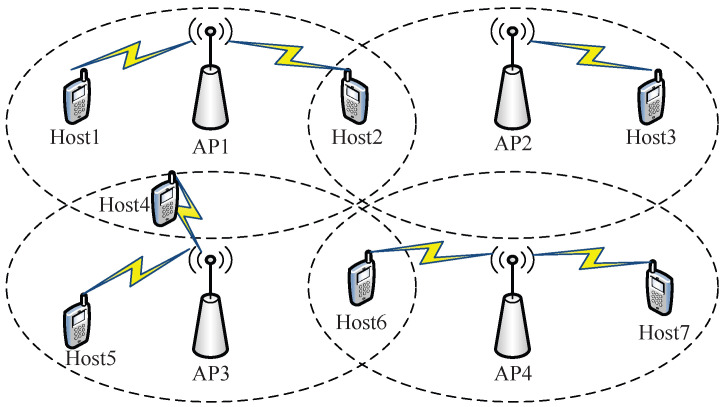
Dense WLAN deployment example.

**Figure 2 sensors-23-06367-f002:**
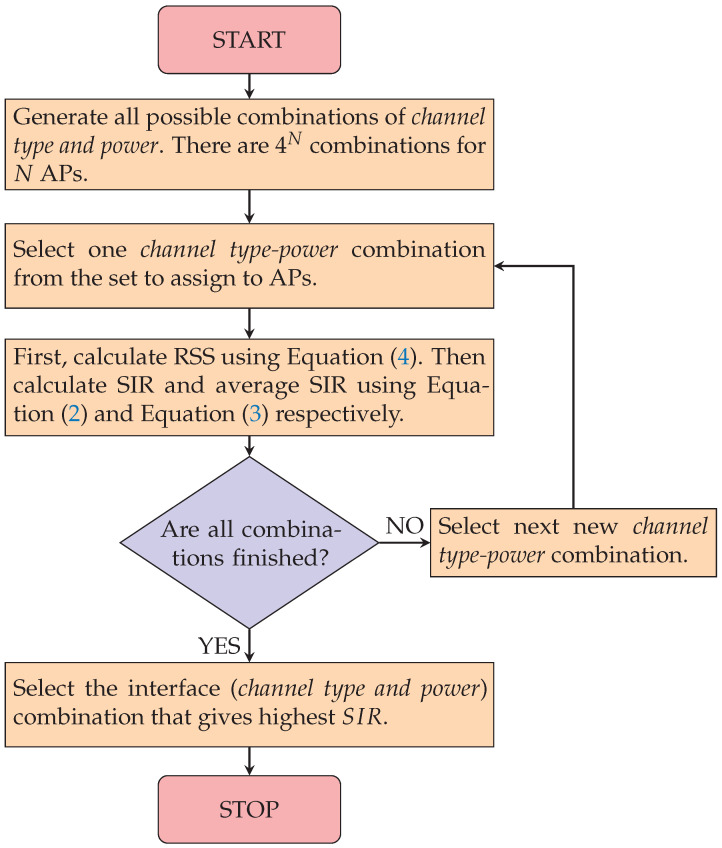
Flow of the AP interface setup optimization method.

**Figure 3 sensors-23-06367-f003:**
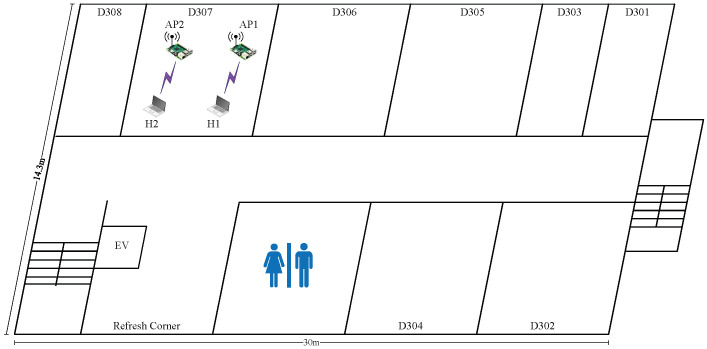
Topology 1 for high interference in *OU-Eng*.

**Figure 4 sensors-23-06367-f004:**
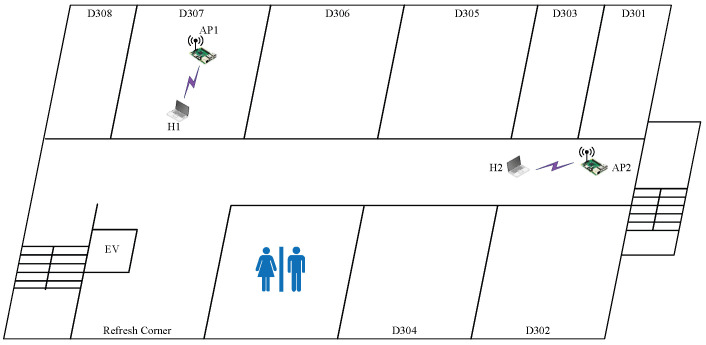
Topology 2 for low interference in *OU-Eng*.

**Figure 5 sensors-23-06367-f005:**
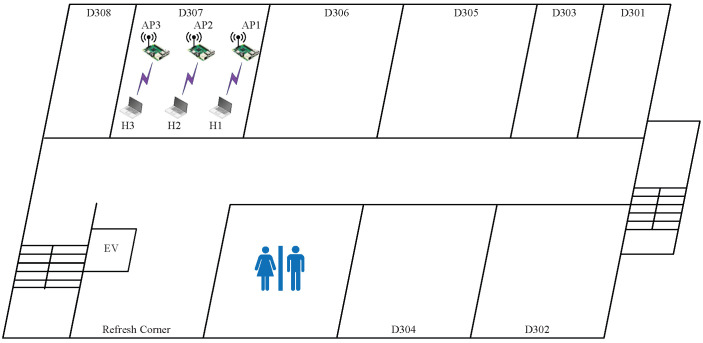
Topology 3 for very high interference in *OU-Eng*.

**Figure 6 sensors-23-06367-f006:**
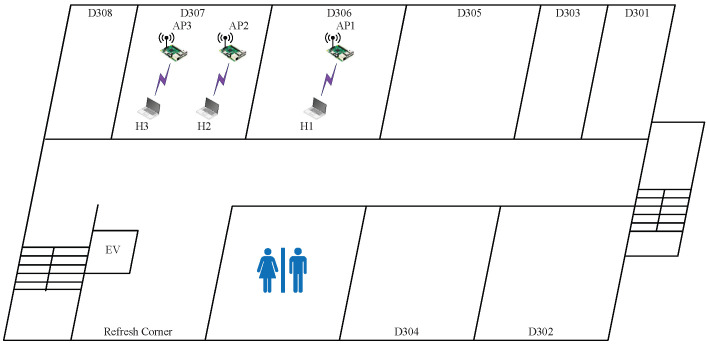
Topology 4 for high interference in *OU-Eng*.

**Figure 7 sensors-23-06367-f007:**
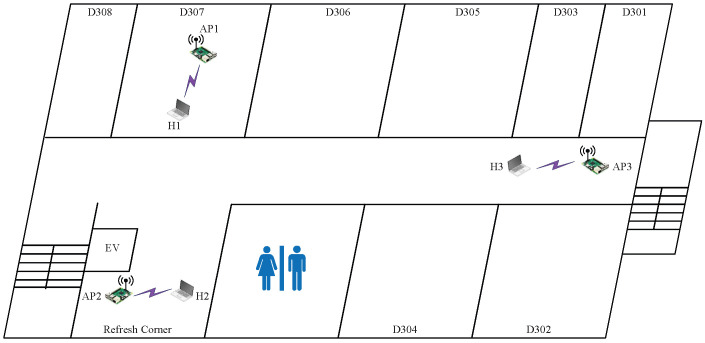
Topology 5 for low interference in *OU-Eng*.

**Figure 8 sensors-23-06367-f008:**
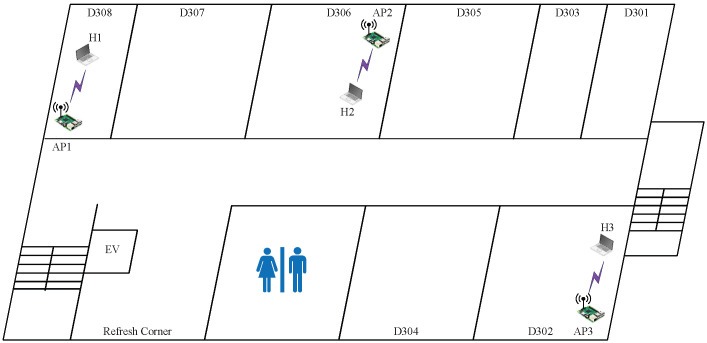
Topology 6 for very low interference in *OU-Eng*.

**Figure 9 sensors-23-06367-f009:**
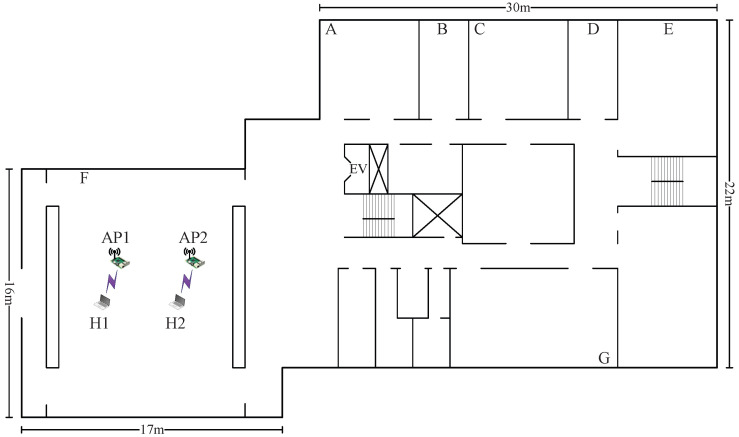
Topology 7 for high interference in *OU-Grad*.

**Figure 10 sensors-23-06367-f010:**
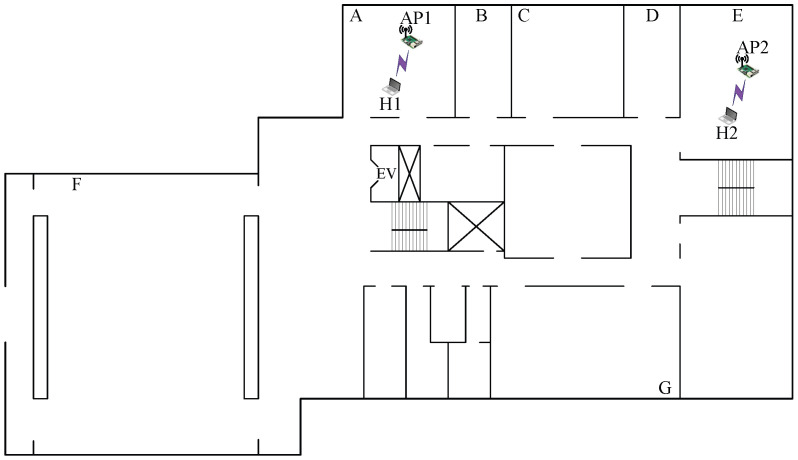
Topology 8 for low interference in *OU-Grad*.

**Figure 11 sensors-23-06367-f011:**
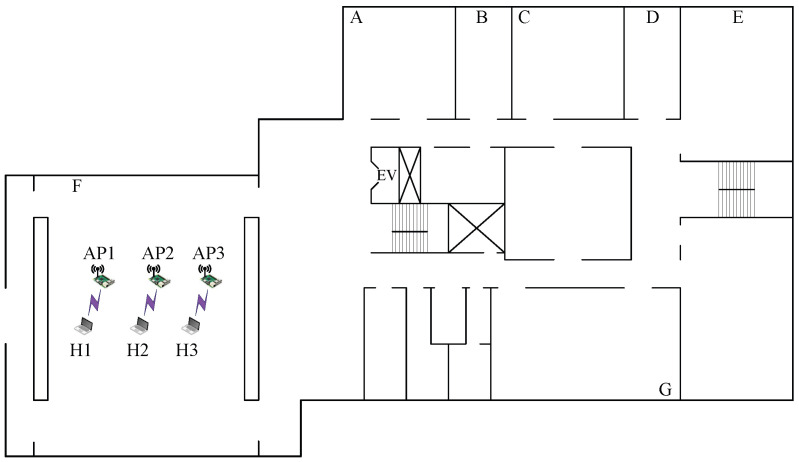
Topology 9 for high interference in *OU-Grad*.

**Figure 12 sensors-23-06367-f012:**
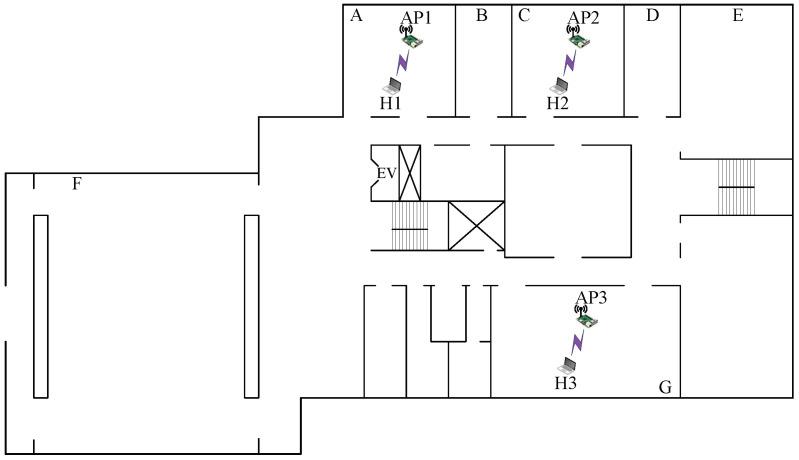
Topology 10 for low interference in *OU-Grad*.

**Figure 13 sensors-23-06367-f013:**
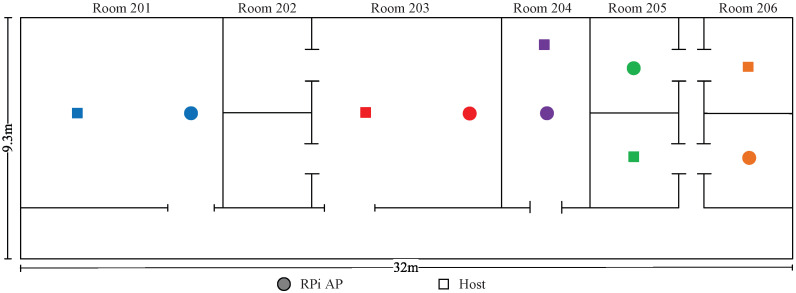
Topology 11 for five APs in *JU-Sci*.

**Figure 14 sensors-23-06367-f014:**
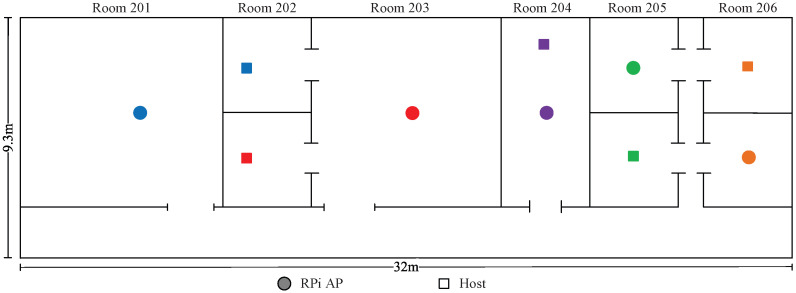
Topology 12 for five APs in *JU-Sci*.

**Figure 15 sensors-23-06367-f015:**
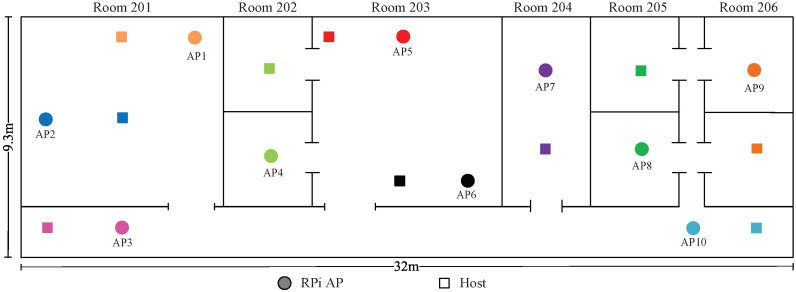
Topology 13 for 10 APs in *JU-Sci*.

**Figure 16 sensors-23-06367-f016:**
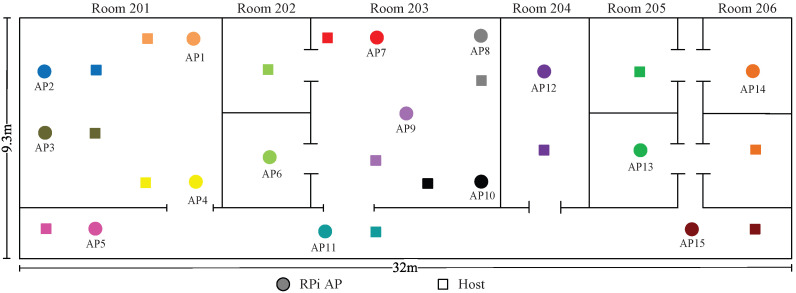
Topology 14 for 15 APs in *JU-Sci*.

**Table 1 sensors-23-06367-t001:** Comparisons of relevant issues between related works and our proposal.

Characteristic		[8]	[13]	[14]	[15]	[16]	[17]	[18]	[19]	[20]	Proposed
Channel assignment	Non-CB	◯	◯	◯	◯	X	◯	◯	◯	◯	◯
	CB	X	X	X	◯	X	◯	X	◯	X	◯
Power assignment		◯	◯	◯	◯	◯	X	◯	◯	◯	◯
Simultaneous implementation		◯	X	X	◯	X	X	◯	◯	◯	◯
Evaluation	Testbed	X	X	X	X	X	X	X	X	X	◯
	Simulation	◯	◯	◯	◯	◯	◯	◯	◯	◯	◯

**Table 2 sensors-23-06367-t002:** PC platform.

Processor	Intel Core i5-4570 CPU @ 3.20 GHz
Memory	8 GB
Operating system	Ubuntu LTS 18.10
Programming language	C++

**Table 3 sensors-23-06367-t003:** Parameters in the throughput estimation model.

Parameter	Value		Note
	Non-CB (20 MHz)	CB (40 MHz)	
${P_{1}}_{max}$	−20 (dBm)	−28.3 (dBm)	Max. transmission power
${P_{1}}_{min}$	−28 (dBm)	−33.2 (dBm)	Min. transmission power
α	2.9	2.9	Path loss exponent
	13	9	Number of channels
	2.4 GHz	2.4 GHz	Frequency
$d_{max}$	90 m	90 m	Covering range of AP
*a*	40.0	55.0	Throughput estimation
*b*	50.50	54.0	Throughput estimation
*c*	6.50	8.05	Throughput estimation

**Table 4 sensors-23-06367-t004:** Devices and software specifications.

AP	Model	Raspberry Pi 3 B+
	CPU	Broadcom BCM2837B0 (1.4 GHz)
	USB NIC	TP-Link TL-WN722N
	Operation mode	IEEE 802.11n, 2.4 GHz
	Channel width	20 MHz/40 MHz
server PC	Model	Fujitsu Lifebook S761/C
	CPU	Intel Core i5-2520M@2.5 Ghz
	RAM	4GB DDR3 1333 MHz
	OS	Ubuntu 14.04 LTS
host PC	Model	Toshiba Dynabook R731/B
	CPU	Intel Core i5-2520M (2.5 GHz)
	RAM	4GB DDR3 1333 MHz
	OS	Ubuntu 14.04 LTS
software	Name	Version
	hostapd [27]	2.9
	iperf [28]	2.05
	iw [29]	5.9

**Table 5 sensors-23-06367-t005:** Device locations.

Network Field	#APs	Topology	Device Locations (${\mathrm{AP}}_{i}$, ${\mathrm{Host}}_{i}$)				
*OU-Eng*	2 APs	1	D307	D307			
		2	D307	corr. near D302			
	3 APs	3	D307	D307	D307		
		4	D307	D307	D306		
		5	D307	refresh corner	corr. near D302		
		6	D308	D306	D302		
*OU-Grad*	2 APs	7	F	F			
		8	F	E			
	3 APs	9	F	F	F		
		10	A	C	G		
*JU-Sci*	5 APs	11	201	203	204	205	206
		12	201, 202	203, 202	204	205	206
	10 APs	13	201	201	corr. near 201	202	203
			203	204	205	206	corr. near 206
	15 APs	14	201	201	201	201	corr. near 201
			202	203	203	203	203
			corr. near 203	204	205	206	corr. near 206

**Table 6 sensors-23-06367-t006:** Results for topology 1 in *OU-Eng*.

Power	0 CB			1 CB			2 CB		
	[1,13]			[1,(9+13)]			[(1+5),(9+13)]		
	e.SIR	a.SIR	t.thp	e.SIR	a.SIR	t.thp	e.SIR	a.SIR	t.thp
H, H	0.39	0.44	74.50	0.41	0.44	76.48	0.42	0.45	79.33
H, L	0.46	0.58	76.54	0.78	0.97	77.53	0.82	1.09	82.75
L, H	0.36	0.62	54.58	0.40	0.76	66.25	0.40	0.84	67.50
L, L	0.41	0.56	54.56	0.41	0.79	60.67	0.42	0.90	61.90

Yellow color in a column presents the highest value for the respective channel-power setup whereas the orange color presents the optimal result among all channel-power combinations. The same notes apply to below tables.

**Table 7 sensors-23-06367-t007:** Results for topology 2 in *OU-Eng*.

Power	0 CB			1 CB			2 CB		
	[1,13]			[1,(9+13)]			[(1+5),(9+13)]		
	e.SIR	a.SIR	t.thp	e.SIR	a.SIR	t.thp	e.SIR	a.SIR	t.thp
H, H	283.38	291.29	84.76	482.74	502.98	85.98	782.30	736	86
H, L	161.45	236.77	97.58	161.45	706.10	107.25	749.41	1325	109
L, H	211.54	437.58	74.58	455.71	441.15	87.88	852.53	445	89
L, L	289.61	487.25	115.25	582.55	916.50	117.5	911.29	1433	119

**Table 8 sensors-23-06367-t008:** Results for topology 3 in *OU-Eng*.

Power	0 CB			1 CB			2 CB			3 CB		
	[1,7,13]			[5,1,(9+13)]			[(1+5),(9+13),13]			[(1+5),(1+5),(9+13)]		
	e.SIR	a.SIR	t.thp	e.SIR	a.SIR	t.thp	e.SIR	a.SIR	t.thp	e.SIR	a.SIR	t.thp
H, H, H	0.64	0.27	101.6	0.65	0.20	69.10	0.61	0.69	66.56	0.41	0.55	47.1
H, H, L	0.57	0.08	92.47	0.57	0.25	61.00	0.54	1.21	81.25	–	–	–
H, L, H	0.78	0.81	104.87	0.69	0.19	60.33	0.69	1.07	77.74	–	–	–
H, L, L	0.71	0.06	88.65	0.71	0.05	57.21	0.71	2.21	84.12	–	–	–
L, H, H	0.61	0.37	90.65	0.61	0.55	78.56	0.61	0.75	65.92	0.62	0.61	54.2
L, H, L	0.57	0.51	93.15	0.57	0.19	73.65	0.55	0.79	67.35	–	–	–
L, L, H	0.67	0.33	85.2	0.61	0.23	64.90	0.61	0.97	72.64	0.63	0.78	56.31
L, L, L	0.63	0.24	82.27	0.63	0.20	63.80	0.63	1.33	63.13	0.70	0.89	58.8

**Table 9 sensors-23-06367-t009:** Results for topology 4 in *OU-Eng*.

Power	0 CB			1 CB			2 CB			3 CB		
	[1,7,13]			[5,1,(9+13)]			[(9+13),(1+5),13]			[(9+13),(1+5),(1+5)]		
	e.SIR	a.SIR	t.thp	e.SIR	a.SIR	t.thp	e.SIR	a.SIR	t.thp	e.SIR	a.SIR	t.thp
H, H, H	12.34	18.34	111.85	17.47	15.42	101.35	8.83	7.85	81.08	7.31	11.624	68.91
H, H, L	9.70	10.61	102.35	9.70	4.73	85.55	8.19	7.53	79.01	–	–	–
H, L, H	13.30	4.34	105.50	17.74	1.14	90.72	8.71	11.28	83.25	–	–	–
H, L, L	9.45	1.69	98.20	9.45	11.40	98.40	6.47	7.74	81.63	–	–	–
L, H, H	15.20	5.11	102.00	18.53	21.86	112.10	14.62	9.54	80.65	7.55	13.03	69.52
L, H, L	12.75	9.86	104.10	12.75	1.02	90.45	14.01	10.75	82.03	–	–	–
L, L, H	15.78	9.76	104.80	17.83	0.32	83.15	15.78	15.44	91.48	9.33	15.54	73.95
L, L, L	12.23	1.56	99.95	12.23	2.49	95.85	12.23	6.15	80.25	12.23	16.305	75.91

**Table 10 sensors-23-06367-t010:** Results for topology 5 in *OU-Eng*.

Power	0 CB			1 CB			2 CB			3 CB		
	[1,7,13]			[5,1,(9+13)]			[(1+5),(9+13),13]			[(1+5),(9+13),(5+9)]		
	e.SIR	a.SIR	t.thp	e.SIR	a.SIR	t.thp	e.SIR	a.SIR	t.thp	e.SIR	a.SIR	t.thp
H, H, H	187.67	372.27	113.25	430.64	23.82	98.64	314.53	186.13	100.93	358.17	101.82	75.4
H, H, L	122.72	330.19	109.25	122.72	11.47	97.84	128.61	87.25	89.75	206.51	142.37	82.89
H, L, H	203.26	103.79	103.50	455.28	333.62	120.80	455.28	200.03	112.55	–	–	–
H, L, L	126.62	46.05	96.90	126.62	16.84	106.10	126.62	192.63	102.58	408.22	236.83	86.3
L, H, H	202.98	21.20	104.40	448.65	62.31	105.80	327.74	344.08	126.80	–	–	–
L, H, L	132.98	26.89	105.20	132.98	18.97	104.53	138.07	126.06	90.16	–	–	–
L, L, H	207.17	60.77	101.43	462.42	412.05	126.10	463.42	441.34	131.30	–	–	–
L, L, L	123.04	64.67	96.55	123.04	94.47	115.00	124.04	102.17	101.00	455.62	348.07	91.1

**Table 11 sensors-23-06367-t011:** Results for topology 6 in *OU-Eng*.

Power	0 CB			1 CB			2 CB			3 CB		
	[1,7,13]			[1,5,(9+13)]			[(1+5),13,(9+13)]			[(1+5),(9+13),(5+9)]		
	e.SIR	a.SIR	t.thp	e.SIR	a.SIR	t.thp	e.SIR	a.SIR	t.thp	e.SIR	a.SIR	t.thp
H, H, H	665.75	632.79	100.46	816.02	606.85	103.17	809.32	674.69	100.60	856.48	881.12	149.40
H, H, L	232.73	704.94	105.65	816.91	689.68	107.90	816.91	731.08	106.60	851.75	701.39	147.10
H, L, H	502.64	687.87	103.73	173.79	688.91	108.87	186.84	706.23	104.75	597.35	620.68	131.00
H, L, L	172.19	690.49	104.60	172.19	756.29	108.57	172.19	434.84	99.40	581.23	372.14	128.26
L, H, H	509.23	760.51	115.87	820.13	449.29	103.80	810.83	624.81	113.40	810.83	637.42	135.50
L, H, L	398.54	712.88	108.25	821.34	837.95	117.30	821.34	850.81	127.80	821.34	341.92	116.80
L, L, H	206.38	741.29	112.87	100.34	795.44	113.65	113.01	788.73	119.06	113.01	626.37	135.03
L, L, L	209.24	679.94	106.97	98.80	754.85	112.17	98.80	372.94	114.40	98.80	395.84	127.80

**Table 12 sensors-23-06367-t012:** Results for topology 7 in *OU-Grad*.

Power	0 CB			1 CB			2 CB		
	[1,13]			[1,(9+13)]			[(1+5),(9+13)]		
	e.SIR	a.SIR	t.thp	e.SIR	a.SIR	t.thp	e.SIR	a.SIR	t.thp
H, H	0.39	1.05	48.27	0.41	1.20	54.65	0.42	1.21	66.25
H, L	0.46	1.32	66.54	0.78	1.39	69.78	0.82	1.45	70.68
L, H	0.36	1.24	58.55	0.40	1.21	63.41	0.40	1.35	63.50
L, L	0.41	0.76	62.19	0.41	0.85	66.67	0.42	0.92	67.70

**Table 13 sensors-23-06367-t013:** Results for topology 8 in *OU-Grad*.

Power	0 CB			1 CB			2 CB		
	[1,13]			[1,(9+13)]			[(1+5),(9+13)]		
	e.SIR	a.SIR	t.thp	e.SIR	a.SIR	t.thp	e.SIR	a.SIR	t.thp
H, H	303.43	319.05	72.79	322.01	337.21	76.01	338.18	370.8	76.15
H, L	438.45	471.35	70.65	536.94	542.88	73.10	536.94	543.9	74.32
L, H	456.19	492.98	79.50	499.25	511.54	83.59	536.22	520.5	85.1
L, L	792.69	521.88	81.01	792.69	557.91	84.54	792.69	572.5	86.1

**Table 14 sensors-23-06367-t014:** Results for topology 9 in *OU-Grad*.

Power	0 CB			1 CB			2 CB			3 CB		
	[1,7,13]			[5,1,(9+13)]			[(1+5),(9+13),13]			[(1+5),(1+5),(9+13)]		
	e.SIR	a.SIR	t.thp	e.SIR	a.SIR	t.thp	e.SIR	a.SIR	t.thp	e.SIR	a.SIR	t.thp
H, H, H	0.64	0.61	61.77	0.65	0.8	71.7	0.61	0.68	68.21	0.41	0.44	65.25
H, H, L	0.57	0.58	53.94	0.57	0.83	64.8	0.54	0.85	80.75	0.5	0.47	62.33
H, L, H	0.78	0.92	98.8	0.69	0.75	67.9	0.69	0.78	72.25	0.64	0.55	64.2
H, L, L	0.71	0.76	66.6	0.71	0.51	58.1	0.71	0.87	81.7	0.63	0.61	54.65
L, H, H	0.61	0.71	65.19	0.61	0.89	89.7	0.61	0.51	67.91	0.62	0.69	59.75
L, H, L	0.57	0.15	56.47	0.57	0.7	69.07	0.55	0.68	68.55	0.56	0.53	63.8
L, L, H	0.67	0.82	95.7	0.61	0.55	79.13	0.61	0.69	74.21	0.63	0.72	52.64
L, L, L	0.63	0.61	62.07	0.63	0.69	65.7	0.63	0.61	60.2	0.7	0.84	73.1

**Table 15 sensors-23-06367-t015:** Results for topology 10 in *OU-Grad*.

Power	0 CB			1 CB			2 CB			3 CB		
	[1,7,13]			[1,(9+13),5]			[13,(1+5),(9+13)]			[(1+5),(5+9),(9+13)]		
	e.SIR	a.SIR	t.thp	e.SIR	a.SIR	t.thp	e.SIR	a.SIR	t.thp	e.SIR	a.SIR	t.thp
H, H, H	723.52	574.49	108.6	801.18	605.91	117.5	807.97	896.6	154.4	839.86	950	129.2
H, H, L	632.61	570.26	117.3	756.74	459.27	109.6	572.63	755.41	146.3	684.11	262.38	127.4
H, L, H	697.98	637.56	114.7	797.98	523.24	129.9	768.5	933.13	149.8	778.92	991.75	138.7
H, L, L	579.55	347.44	109.6	679.55	700.4	124.8	243.85	207.72	134.4	336.6	979.49	135.3
L, H, H	793.1	736.11	122.3	816.99	713.63	121.8	993.72	1340.1	160.9	929.17	673.06	142.6
L, H, L	725.14	438.55	104.9	770.27	643.36	119.7	728.55	638.7	139.8	770.27	211.59	140.5
L, L, H	790.4	519.82	108.9	890.4	816.02	138.4	954.74	1054.45	149.6	930.72	1052.38	145.8
L, L, L	746.3	566.55	104.7	846.3	671.37	127.3	807.26	726.25	140.1	846.3	1009.37	144.2

**Table 16 sensors-23-06367-t016:** Results for five APs in *JU-Sci*.

Topology		Channel Type	Channel	Power	*e.SIR*
Topology 11	AP1	CB	(1 + 5)	H	6.12
	AP2	non-CB	6	L	
	AP3	non-CB	13	L	
	AP4	non-CB	1	L	
	AP5	CB	(9 + 13)	H	
Topology 12	AP1	CB	(9 + 13)	H	9.25
	AP2	CB	(1 + 5)	L	
	AP3	non-CB	13	H	
	AP4	CB	(7 + 11)	L	
	AP5	non-CB	1	H	

**Table 17 sensors-23-06367-t017:** Results for 10 APs in *JU-Sci*.

Topology		Channel Type	Channel	Power	*e.SIR*
Topology 13	AP1	non-CB	1	H	12.97
	AP2	non-CB	5	L	
	AP3	CB	(9 + 13)	H	
	AP4	non-CB	7	H	
	AP5	non-CB	11	L	
	AP6	non-CB	3	L	
	AP7	CB	(1 + 5)	L	
	AP8	non-CB	13	L	
	AP9	non-CB	1	H	
	AP10	CB	(5 + 9)	H	

**Table 18 sensors-23-06367-t018:** Results for 15 APs in *JU-Sci*.

Topology		Channel Type	Channel	Power	*e.SIR*
Topology 14	AP1	non-CB	1	H	10.24
	AP2	non-CB	5	L	
	AP3	non-CB	9	L	
	AP4	non-CB	13	L	
	AP5	CB	(1 + 5)	H	
	AP6	non-CB	7	H	
	AP7	non-CB	11	L	
	AP8	non-CB	1	L	
	AP9	non-CB	5	L	
	AP10	non-CB	13	L	
	AP11	non-CB	10	H	
	AP12	non-CB	9	L	
	AP13	non-CB	1	L	
	AP14	non-CB	5	H	
	AP15	CB	(9 + 13)	H	

**Table 19 sensors-23-06367-t019:** Summary of overall experimental results.

Network Field	Topology	#APs/ Hosts	#Rooms	Interference Level	Best (Channel Type and Power)
*OU-Eng*	1	2/2	1	high	(CB, H), (CB, L)
	2		2	low	(CB, L), (CB, L)
	3	3/3	1	very high	(non-CB, H), (non-CB, L), (non-CB, H)
	4		2	high	(non-CB, L), (non-CB, H), (CB, H)
	5		3	low	(CB, L), (CB, L), (non-CB, H)
	6		3	very low	(CB, H), (CB, H), (CB, H)
*OU-Grad*	7	2/2	1	high	(CB, H), (CB, L)
	8		2	low	(CB, L), (CB, L)
	9	3/3	1	high	(non-CB, H), (non-CB, L), (non-CB, H)
	10		3	low	(non-CB, L), (CB, H), (CB, H)
*JU-Sci*	11	5/5	5	moderate	(CB, H), (non-CB, L), (non-CB, L), (non-CB, L), (CB, H)
	12		6	moderate	(CB, H), (CB, L), (non-CB, H), (CB, L), (non-CB, H)
	13	10/10	6	high	(non-CB, H), (non-CB, L), (CB, H), (non-CB, H), (non-CB, L), (non-CB, L), (CB, L), (non-CB, L), (non-CB, H), (CB, H)
	14	15/15	6	very high	(non-CB, H), (non-CB, L), (non-CB, L), (non-CB, L), (CB, H), (non-CB, H), (non-CB, L), (non-CB, L), (non-CB, L), (non-CB, L), (non-CB, H), (non-CB, L), (non-CB, L), (non-CB, H), (CB, H)

## Data Availability

Not applicable.

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
