# Peer review of "An Interface Setup Optimization Method Using a Throughput Estimation Model for Concurrently Communicating Access Points in a Wireless Local Area Network"

_sensors, 2023, doi:10.3390/s23146367_

Round 1

Reviewer 1 Report

Dear Author(s),

       I have reviewed the paper, the following are some suggestions- 

1. Discuss the figure 11 in the manuscript. 

2. All the mathematical equations must be cited and discussed properly.

3. The authors must compare the proposed model with the existing techniques.

4. The authors must discuss the novelty of the proposed model.

5. The abstract should be more effective.

Thank you.

Author Response

Respected reviewer,

Thank you for your time and efforts to review our paper. We tried our best to address all your concerns. Please find the attachment for details.

With regards,
Fatema Akhter
Okayama University

Reviewer 2 Report

This work is correct. The biggest criticism that I have, is limited to 3 AP. It will be interesting if the method could be expanded to a bigger number of AP and if it is possible to develop a solution that could be implemented in the actual AP trying to solve the actual chaotic situation.

There are several small problems, in my opinion, in the text.

In line 146, Hi has not been defined (It is easy to deduct that it is the Hosti)

In page 6, expression 6, it is the same as expression 1, this can be removed.

In Table 1 c++ is not a program, it is a language. The program is the compiler, and the program, and the different software used (hostapd, iperf ..) uses c as language.

The figures that represent the topology, should include a scale, this can help to estimate the distances.

The identifications used "Eng. Bldg. #2 OU", "Grad Sch Bldg. OU." are a bit confusing, and they are used in many part of the text.

Probably is more simple to define Scenario 1, Scenario 2 ... and include a table describing every one of the scenarios.

Author Response

(The authors gave the same response as above.)

Reviewer 3 Report

The IEEE 802.11 wireless local-area network (WLAN) is widely used worldwide. However, dense WLAN networks can experience performance issues due to high radio interferences caused by neighboring access points (APs). To tackle this problem, this study was conducted on optimizing AP transmission power to maximize the average signal-to-interference ratio (SIR) among concurrently communicating APs.

The authors claimed that the existing methods are impractical and time-consuming as the number of APs increased because of the requirement of measuring the receiving signal strength (RSS) for all possible combinations of power settings. Moreover, the existing methods do not consider the potential benefits of non-CB in high-interference scenarios.

To address these limitations, the researchers proposed an AP interface setup optimization method that utilized a throughput estimation model for concurrently communicating APs. This method not only selected the maximum or minimum power for each AP but also determined whether to use CB or non-CB. By incorporating this model, the need for expensive RSS measurements under numerous combinations was avoided.

To evaluate the proposed method, extensive experiments were conducted by using Raspberry Pi for APs and Linux PCs for hosts in 12 network topologies across three real buildings. The results confirmed that the proposed method successfully identified the optimal AP interface setup, resulting in the highest total throughput in any given topology.

This manuscript is overall easy to read. The proposed method seems effective because of its simplicity. However, the performance evaluation section cannot demonstrate its advantage against the previous methods. First, there are only up to five APs included in the experiments. As compared to real environments with tens or more APs, the experiments should be enlarged to show its effectiveness. Second, the setting of max/min transmission power should be loosen by including more different power level. Practically, the modern APs can support many different level transmission power. It is not reasonable to ignore the flexibility of APs. Third, the authors should correlate the description of references in Section 2 based on their methods and/or contributions. Fourth, Section 6-8 describe the performance of the proposed method for different buidings and network topologies. Although the authors attempt to show numerous experiment results, these results could be expressed in an organized and summarized manner. For example, the authors may use low/mid/high interference scenarios to show their results.

There are some typos in the manuscript, but the overall presentation is acceptable. 

Author Response

(The authors gave the same response as above.)

Round 2

Reviewer 1 Report

Dear Author(s),

        The paper is good.

It is good.

Reviewer 3 Report

The revised manuscript has addressed my previous concerns. I suggest to accept it in its current form.